# Inter-Strain Epigenomic Profiling Reveals a Candidate IAP Master Copy in C3H Mice

**DOI:** 10.3390/v12070783

**Published:** 2020-07-21

**Authors:** Rita Rebollo, Mariana Galvão-Ferrarini, Liane Gagnier, Ying Zhang, Ardian Ferraj, Christine R. Beck, Matthew C. Lorincz, Dixie L. Mager

**Affiliations:** 1Terry Fox Laboratory, British Columbia Cancer, Vancouver, BC V5Z1L3, Canada; liane.gagnier@bccdc.ca (L.G.); ying.zhang.phd@gmail.com (Y.Z.); 2University of Lyon, INSA-Lyon, INRA, BF2i, UMR0203, F-69621 Villeurbanne, France; mariana.galvao-ferrarini@insa-lyon.fr; 3Department of Genetics and Genome Sciences, University of Connecticut Health Center, Farmington, CT 06030, USA; Ardian.Ferraj@jax.org (A.F.); Christine.Beck@jax.org (C.R.B.); 4The Jackson Laboratory for Genomic Medicine, Farmington, CT 06032, USA; 5Department of Medical Genetics, University of British Columbia, Vancouver, BC V6T1Z3, Canada; mlorincz@mail.ubc.ca

**Keywords:** transposable elements, endogenous retrovirus, mouse, C3H, IAP, epigenetics

## Abstract

Insertions of endogenous retroviruses cause a significant fraction of mutations in inbred mice but not all strains are equally susceptible. Notably, most new Intracisternal A particle (IAP) ERV mutagenic insertions have occurred in C3H mice. We show here that strain-specific insertional polymorphic IAPs accumulate faster in C3H/HeJ mice, relative to other sequenced strains, and that IAP transcript levels are higher in C3H/HeJ embryonic stem (ES) cells compared to other ES cells. To investigate the mechanism for high IAP activity in C3H mice, we identified 61 IAP copies in C3H/HeJ ES cells enriched with H3K4me3 (a mark of active promoters) and, among those tested, all are unmethylated in C3H/HeJ ES cells. Notably, 13 of the 61 are specific to C3H/HeJ and are members of the non-autonomous 1Δ1 IAP subfamily that is responsible for nearly all new insertions in C3H. One copy is full length with intact open reading frames and hence potentially capable of providing proteins in *trans* to other 1Δ1 elements. This potential “master copy” is present in other strains, including 129, but its 5’ long terminal repeat (LTR) is methylated in 129 ES cells. Thus, the unusual IAP activity in C3H may be due to reduced epigenetic repression coupled with the presence of a master copy.

## 1. Introduction

Transposable elements (TEs) are repeated DNA sequences that are generally grouped into two classes depending on their transposition mechanism: retrotransposons copy themselves within the genome through an RNA intermediate, while DNA elements transpose directly via a DNA molecule. TEs have been observed in nearly all genomes analyzed to date, attesting to their effectiveness at colonizing host genomes and contributing to genome size variation and diversity between and within species [1,2]. Furthermore, TEs spread genome-wide regulatory sequences present within their copies, allowing for the establishment of gene regulatory networks and playing an important role in host genetic programs [3,4]. In humans, where TE sequences represent around half the genome, relatively few TE mutations have been described, compared to other types of mutations, and all of them are due to insertions of L1 Long interspersed elements (LINEs) or Short Interspersed Elements (SINEs) [5]. In mice, however, TEs are a major ongoing mutagenic source, responsible for up to 14–16% of all published germ-line mutations in inbred strains [6,7,8]. Interestingly, the majority of mouse TE insertional mutations are caused by endogenous retroviruses (ERVs), which are signatures of past retroviral infections of the germ-line that integrated into the host genome and adopted an endogenous life-cycle [7,8]. Around 10% of the mouse genome is composed of ERV sequences of many different families [6,9]. Intracisternal A-particle sequences (IAP) [10], a particular ERV family, is responsible for 40% of all described TE mutations in inbred mice, as tabulated in our recent review [8]. Thousands of IAP copies are found in the mouse genome, and while most of them are non-autonomous, around 600 appear full-length in the reference strain C57BL/6 and have identical long terminal repeats (LTRs), suggesting recent transposition [11].

To restrict harmful TE activity, various cellular proteins and epigenetic mechanisms are employed by host cells to silence ERVs and other TEs [2,12,13,14,15]. IAP elements in particular are generally heavily methylated [16,17,18] and the target of repressive histone modifications, such as H3K9me3 deposited by the SETDB1/KAP1 complex [13,19,20], through Krüppel-associated box zinc finger proteins (KRAB-ZFPs) [21]. The loss of these epigenetic repressive marks causes the activation of IAP transcription [16,19,22] and can lead to somatic retrotransposition and oncogene activation in mouse cancer models [23]. Interestingly, 46 IAP insertional mutations have been described in mice and the strain of origin has been documented for 43 of them. Among these 43 cases, 36 have occurred in C3H/HeJ mice, other C3H substrains or in a hybrid involving C3H, while only seven have been identified in other mouse strains, as shown in Figure 1A (adapted from [8]). This extreme strain bias suggests that the evasion of host silencing and the retrotransposition of IAPs occurs nearly exclusively in C3H mice. Moreover, nearly all new IAP-induced mutations have been caused by insertions of a particular non-autonomous, partly deleted subfamily, termed 1Δ1 [7,8]. As further evidence of the retrotranspositional activity of this subfamily in C3H mice, it has also been shown that at least 26 1Δ1 IAP insertions, present in C3H/HeJ, are absent from the highly related C3HeB/FeJ substrain [24]. This IAP subfamily has a deletion in the *gag-pol* region, which creates a GAG-POL fusion protein that facilitates retrotransposition in *cis*, but it still requires a source of GAG and POL in *trans* from autonomous full-length IAPs in order to retrotranspose [25]. 

In this study, we investigated possible mechanisms for the extraordinary IAP activity in C3H mice. We took advantage of sequences flanking IAP elements to identify specific copies located within permissive chromatin regions in C3H/HeJ embryonic stem (ES) cells. We uncovered 61 IAP copies associated with the permissive mark H3K4me3, among which only one copy is potentially capable of generating the full complement of retrotransposition machinery. This potential “master copy” is also present in other strains, including 129, but its chromatin state differs between C3H/HeJ and 129-derived ES cells. Based on these observations, we propose that the unusually high activity of IAP in C3H mice may be due to a combination of copy specific insertions and reduced epigenetic repression.

## 2. Materials and Methods 

### 2.1. Biological Material

C3H/HeJ mES cells (Jackson laboratory, C3H/HeJ-PRX-C3H #2) were cultured on irradiated feeders (CF-1 mouse embryonic fibroblasts, MEFs), and maintained on mouse ES cell media and passaged every day. TT2 (C57BL/6 x CBA F1), J1 (129S4/SvJae) and C2 (C57BL/6NTac) ES cells were cultured on mouse ES cell media on gelatinized plates and passaged every 48–72 h, as previously described [20]. Mouse ES cell media is composed of DMEM, supplemented with 15% FBS (HyClone), 20 mM HEPES, 0.1 mM nonessential amino acids, 0.1 mM 2-mercaptoethanol, 100 units/mL penicillin, 0.05 mM streptomycin, leukemia inhibitory factor (LIF) and 2 mM glutamine. MEFs were cultured on DMEM, supplemented with 10% FBS, and irradiated at early passages. For all downstream analyses, C3H/HeJ ES cells were depleted of feeders before pelleting.

### 2.2. Transposable Element Copies and SNP Analysis 

IAP private copy numbers and SNP private numbers were retrieved from previously published studies [26,27]. The statistical analysis of private IAP copies versus private SNPs was performed on GraphPad prism by using a linear regression model with 95% confidence interval, by calculating the cook distance (0.7403), which is superior to the 0.4 threshold suggesting C3H/HeJ is an influential outlier and, finally, by computing the standard deviation of the residues with and without the C3H/HeJ datapoints, showing a decrease by half in the absence of C3H/HeJ. All statistical analyses are in agreement with C3H/HeJ, having accumulated more IAP copies than the other strains, as shown in Appendix A.

IAP insertional polymorphism information was obtained from a previous study [26] and manual annotation and presence in other strains of H3K4me3 C3H/HeJ copies were analysed using the Mouse strain assembly hub (http://hgdownload.soe.ucsc.edu/hubs/mouseStrains/hubIndex.html) [28]. Finally, C3H/HeJ IAP copies were retrieved from the C3H/HeJ genome (Accession: GCA_001632575.1/C3H_HeJ_v1). 

### 2.3. IAP Expression Analysis

Total RNA was extracted (two biological replicates for each ES cell line) with the All Prep DNA/RNA mini kit from Qiagen (Hilden, Germany). RNA was treated with the Turbo DNA-free kit from Ambion (Austin, TX, USA) in order to remove DNA. Then 1 µg total RNA extracts was reverse transcribed with SuperScript II reverse transcriptase system [Invitrogen (Carlsbad, CA, USA)]. We synthesized two different cDNAs (65 °C for 5 min, 25 °C for 5 min, 50 °C for 60 min and 70 °C for 15 min)—a control reaction with no reverse transcriptase to test for DNA contamination and a pool of total cDNA synthesized with random primers. The cDNA samples were diluted 10-fold and PCR was carried out using Fast SYBR Green Master Mix [Applied Biosystems (Foster City, CA, USA)] using specific primers for each gene analyzed. Primers were chosen overlapping the internal deletion found in 1Δ1 copies for full-length copies, surrounding the deletion for 1Δ1 copies and primers downstream of the deletion within the *pol* gene for total IAP copies. The quantitative PCR cycling conditions were 20 s at 95 °C (1 cycle) and then 3 s at 95 °C, followed by 30 s at 60 °C (45 cycles). Reactions were done in duplicate and standard curves were calculated from serial dilutions of cDNA. The quantity of the transcripts was estimated relative to the expression of *tubulin*, *actin* and *TBP* (TATA binding protein), chosen as the most stable genes out of six reference genes tested using the GeNorm method [29] with the equation ‘‘Absolute quantity = ‘‘Efficiency of primersˆ (−Ct)’’. Primer efficiencies were equivalent and chosen between 1.9 and 2. All primers are listed in Appendix A. GraphPad Prism was used for data visualization and statistical testing (Anova with Dunnett’s pos-hoc test).

### 2.4. Native ChIP-Seq

Chromatin immunoprecipitation was performed as previously described [19]. Ten million TT2 and feeder depleted C3H/HeJ cells were harvested for nChIP-seq using antibodies against H3K4me3 (Abcam ab1012, 5 µL) and H3K9me3 (Active Motif 39161, 5 μL). Illumina library construction was performed as described in [30]. Paired-end sequencing (100 bp) was performed on an Illumina HiSeq 2000. Data from this study have been submitted to GEO (GSE152208).

Raw reads were filtered for low quality, and adapter sequences were trimmed with Trimmomatic [31]. Clean reads were mapped against the reference genome C57BL/6J, (mm10, Accession: GCA_000001635.6/GRCm38.p4) and the draft C3H/HeJ genome (Accession: GCA_001632575.1/C3H_HeJ_v1) with Bowtie2 [32], with an overall alignment from 90 to 95%. Multiple mapped reads or reads that aligned with more than two mismatches were excluded from subsequent analyses. Peak calling was performed for each bowtie2 bam file against their controls using MACS2 software [33] and an FDR cut-off of 0.01. Bedtools [34] was used to intersect the IAP copies (±250 bp) with the narrow peaks from the previous analyses.

### 2.5. Other Data Sets

Peaks and bigwig of H3K4me3 datasets for ModEncode Bruce4 ES cells (C57BL/6, GSM769008) and E14 ES cells (129/Ola, GSM1000124) [35] were downloaded from the new Cistrome data browser [36]. Bedtools [34] was used to intersect the IAP copies (± 250 bp) with the narrow peaks from the downloaded datasets. 

### 2.6. Enrichment Profiles

The enrichment profiles were generated from bigwig files using deeptools suite [37] and ggplot2 package [38] from R. We estimated H3K4me3 enrichment around gene transcriptional start sites of all the datasets studied, as shown in Appendix A, by using the computeMatrix function from deeptools with reference-point and the following parameters: --referencePoint TSS -a 3000 -b 3000 -bs 50. We also compared the enrichment of H3K4me3 or H3K9me3 around the flanking regions of IAP copies (i) in C3H/HeJ, absent in C3H/HeJ but present either in (ii) C57BL/6, (iii) 129 or (iv) common to all strains. For this, we used the computeMatrix function with scale-regions mode with the following parameters: -b 3000 -a 3000 --bs 1 -m 1.

### 2.7. Native ChIP-qPCR

ChIP-qPCR was performed as previously described [20]. Ten million C3H/HeJ feeder depleted cells were pelleted in duplicates and antibody references and quantities were equivalent to the nChIP-seq method. Immunoprecipitated DNA was quantified by using the PicoGreen system from Invitrogen. Then, 0.05 ng/µL of ChIP material was analyzed in technical duplicates through quantitative PCR (Fast SYBR Green Master Mix from Applied Biosystems) by comparing the amplification of Input DNA relative to immunoprecipitated DNA (IP) using the formula “Efficiency of primers^(CtInput – CtIP)^”, where the efficiency is calculated through serial dilutions of Input DNA (primers efficiencies were all between 1.9 and 2.1).The primers are available in Appendix A.

### 2.8. Bisulfite Analysis of IAP Copies and Their Flanking Sequences

Bisulfite conversion PCR, cloning and sequencing were carried out as described previously [39]. All the sequences included in the analysis either displayed unique methylation patterns or unique C to T non-conversion errors after the bisulfite treatment of the genomic DNA. This avoids considering several PCR-amplified sequences resulting from the same template molecule. All sequences had a conversion rate greater than 95%. Sequences were analyzed with the Quma free online software (RIKEN, Kobe, Japan) [40]. Primers are listed in Appendix A.

### 2.9. Size Determination of C3H/HeJ H3K4me3-IAP Copies

C3H PacBio long reads were aligned to the mm10 reference genome using NGMLR (v0.2.7) [41]. Insertions were called computationally by structural variant callers pbsv (v2.2.2) (https://github.com/PacificBiosciences/pbsv) [42] and Sniffles (v1.0.11) [41]. The size estimation of C3H copies absent from the C57BL/6 genome was conducted by intersecting IAP coordinates with PacBio-derived insertions at the corresponding locations using Bedtools (v2.29.2) [34]. PCR amplification was also performed to confirm IAP copy sizes—the primers are available in Appendix A. 

The chrX potential master copy was sequenced by genome walking, using primers available in Appendix A. The sequence was deposited in GenBank (accession MT559331) and is also present in Appendix A (along with GAG, PRO and POL amino acid sequences). The orthologous copy is also found in the 129 genome and has been previously sequenced [43].

## 3. Results

### 3.1. IAP Accumulation in Mouse Strains

A previous comprehensive analysis of IAP content in the reference mouse strain C57BL/6 reported a total copy number of 5872, which includes all subfamilies of various ages as well as ~3000 solitary LTRs [11]. Among the young IAP subfamilies, over half are insertional polymorphisms [26,44]. Sequence comparisons of 12 laboratory strains revealed that 93 IAP copies in C3H/HeJ are absent from all other strains [26]. However, although 84% of described IAP mutations occurred in C3H or in a C3H background, as shown in Figure 1A, the accumulation of IAP copies, specifically in the C3H/HeJ genome, is not obvious, as other strains harbor as many or more IAP private copies, as shown in Figure 1B. Nonetheless, if IAP elements have been particularly active in C3H mice since strain divergence, private IAP copies that are not highly detrimental would accumulate at a faster rate in C3H, compared to other strains. We mined genomic sequence data available on the 12 mouse laboratory strains [26,27] and compared the accumulation of single nucleotide polymorphisms (SNPs) specific to a single strain with the number of strain-specific IAP copies, as shown in Figure 1C. The SNP pattern reflects general divergence between the strains and, if the rate of retrotransposition has been similar across strains, one would expect the numbers of strain-specific TE insertions to closely correlate with strain-specific SNPs. However, the number of IAPs observed only in C3H/HeJ is significantly higher than expected, based on the strain-specific SNP pattern, as shown in Figure 1C, and the standard deviation of residuals can be seen in Appendix A. Such bias is not observed for another well-known active ERV, ETn/MusD, that shows a less marked strain-specific bias in A/J mice, as shown in Appendix A, as noted previously for mutagenic insertions [7,8]. Furthermore, the pattern of strain-specific L1 LINE insertions closely follows the SNP pattern, as shown in Figure 1C, suggesting that L1 is not unusually active in any of the commonly used laboratory strains. Such findings support the premise that IAP elements, in particular, have been unusually active in C3H/HeJ mice since the derivation and separation of inbred laboratory strains. However, it should be noted that the vast majority of the thousands of IAP copies in the genomes of inbred mice were inserted prior to the separation of the strains. 

### 3.2. IAP Expression in Mouse ES Cells

The earliest reported IAP-induced mutations in C3H mice occurred in the 1950s [8], but the most recent reported case occurred in 2014 [45], revealing that such elements are still active in this strain. We analyzed IAP transcript steady-state levels in a panel of ES cells derived from C3H/HeJ, C57BL/6 and 129 mice, as shown in Figure 2. Specifically, we determined the transcript levels of 1Δ1 and full-length (those containing intact *gag* and *pol* genes) IAPs in a panel of ES cells from different strains. IAP transcript levels are significantly higher in C3H/HeJ ES cells compared to the other strains, for both putative full-length copies and the 1Δ1 subfamily. Such findings are in agreement with IAP expression levels observed in C3H/He tissues compared to C57BL/6 and STS/A mice [46]. Notably, levels of 1Δ1 transcript are 30-fold higher than those of putative full-length copies, as shown in Figure 2, and, as mentioned above, are known to accumulate in C3H sub-strains [24]. Indeed, 1Δ1 copies are preferentially transposed over IAP full-length elements [25]. Nevertheless, 1Δ1 cannot transpose autonomously and hence depends on the transposition machinery of other IAP copies.

### 3.3. Permissive IAP Copies in C3H/HeJ ES Cells

Transcription start sites of canonical IAP RNAs occur within the 5’ LTR, while polyadenylation occurs within the 3’LTR. As flanking sequences in IAP transcripts are lacking, it is not possible to discriminate between highly similar active and silent copies in transcriptomic datasets. However, we and other groups have previously demonstrated that IAP LTRs, serving as gene alternative promoters, are associated with permissive histone marks (H3K4me3) and are hypomethylated [17,47,48]. Hence, in order to pinpoint potential transcriptionally active IAP copies in the C3H genome, we searched for H3K4me3 enrichment at IAP flanking sites (+/− 250 bp) in C3H/HeJ ES cells. In order to maximize the chances of finding IAP permissive copies, we first mapped the H3K4me3 reads against the reference genome (C57BL/6) and searched for enriched peaks in the flanking sequences of previously described IAP copies present in the C3H/HeJ genome [26]. We also mapped the H3K4me3 reads against the C3H/HeJ draft genome and searched for enriched peaks in the flanking sequences of annotated IAP copies. We uncovered 61 IAP copies associated with H3K4me3 chromatin, as shown in Appendix A. Note that all copies in Appendix A have been given an identifier number that we will refer to throughout the manuscript. For all copies absent from the reference genome, we estimated their size by intersecting IAP coordinates with PacBio-derived insertions specific to C3H, and validated estimates with PCR amplification (detailed in the methods section). Two copies of unknown remain size. The majority of identified copies are annotated as belonging to the young IAP subfamilies (EY, LTR1 and LTR2), as shown in Figure 3A, with the caveat that the annotation of copies absent from the reference genome relies on the C3H/HeJ preliminary genome annotation. While most of the detected H3K4me3 IAP elements are solo LTRs (43 copies), as shown in Figure 3B, 13 copies found in a permissive state are the same size as 1Δ1 copies (~5.2 Kb). Notably, 12 of these are absent from the other laboratory strains, as shown in Appendix A, potentially explaining the high expression of 1Δ1 IAP copies observed in C3H/HeJ ES cells, as shown in Figure 2. Finally, only one copy is considered full-length—a 7.2 kb IAP LTR1 element. 

Since the repressive chromatin mark (H3K9me3) is enriched on IAP elements in ES cells of other strains [13], we measured this mark on IAPs in C3H/HeJ ES cells to investigate if a general lack of IAP repression is evident. We performed H3K9me3 ChIP-seq in C3H/HeJ ES cells and found high enrichment in IAP flanking sequences compared to H3K4me3, as shown in Figure 3C. We confirmed that potential full-length or 1Δ1 copies are indeed the target of H3K9me3 by performing H3K9me3 ChIP-qPCR on LTR-int regions, as shown in Appendix A. Hence, the C3H/HeJ genome is capable of targeting IAP copies with the repressive H3K9me3 mark, but at least 61 copies are able to potentially escape host silencing, as shown in Appendix A and Figure 3D, for representative genome browser views.

In order to confirm that H3K4me3 enrichment in IAP flanking sequences is a surrogate for permissive IAP copies in agreement with previous studies [17,47,48], we determined the DNA methylation status of four IAP copies enriched for H3K4me3 in C3H/HeJ ES cells, as shown in Figure 3E, namely the potential full-length copy (IAP #60), two 1Δ1 elements (IAP #30 and #28) and a solitary LTR (IAP #44). All copies were hypomethylated, confirming their permissive states. In addition, CpGs flanking the IAP copies are also hypomethylated, reinforcing the hypothesis that sequences flanking permissive IAPs are marked by a permissive chromatin structure. Comparison of the DNA methylation of both LTRs in two 1Δ1 H3K4me3-marked copies revealed that the 3’LTR sequences were hypermethylated compared to 5’LTRs, as shown in Figure 3E. In contrast, we previously demonstrated that IAP 5’LTRs of repressed elements remain hypermethylated even if close to permissive regions, while 3’LTR methylation is more relaxed [17].

### 3.4. IAP Copies Are Able to Recruit Permissive Chromatin

As a subset of the 61 H3K4me3-IAPs is located near CpG islands or gene transcription start sites (TSSs), as shown in Appendix A, we wondered if H3K4me3 domains, typically associated with such features, influenced the chromatin state of these IAPs. Hence, we searched for H3K4me3 enriched empty sites (i.e., sites devoid of such copies), in other mouse strains. We performed H3K4me3 ChIP-seq in TT2 cells (C57BL/6 × CBA), and also took advantage of ModEncode and analyzed C57BL/6 (Bruce4) and 129/Ola (E14), H3K4me3 datasets. Overall, empty sites are devoid of H3K4me3, as shown in Figure 4A and Appendix A, for ModEncode data and Figure 4B for example browser views. More specifically, there are 27 copies absent from the C57BL/6 genome, but only four empty sites are enriched in H3K4me3 in Bruce4, and three in TT2. For 129/Ola, out of 28 empty sites, only five are in a permissive chromatin state. Most of these few H3K4me3-enriched empty sites were within 2 kb of a CpG Island promoter, or genic TSS, suggesting that, in these cases, the permissive chromatin of the nearby gene likely spread to the IAP inserted in the C3H/HeJ genome, as shown in Figure 4C, for example. Overall, however, empty sites are generally devoid of H3K4me3, indicating that IAP copies are able to recruit H3K4me3-permissive chromatin.

We next measured DNA methylation at the empty sites of two H3K4me3-IAPs (IAP #60 and IAP #28, present in C3H/HeJ and absent from C57BL/6). CpGs flanking the insertion site of IAP copies are hypermethylated in TT2 ES cells, as shown in Figure 4D, while hypomethylated in C3H/HeJ. Such data suggest not only that IAP copies are able to recruit permissive chromatin, but such a permissive state can also spread to nearby sequences.

### 3.5. Difference in Chromatin State of Specific IAP Copies between Mouse Strains

Forty H3K4me3 IAP copies in C3H/HeJ are also present in other mouse strains, as shown in Appendix A. These include the single full-length “master” copy on the X chromosome (IAP #60), as detailed in the following section). A comparison of H3K4me3 in the flanking sequences of common copies between all ES cells studied (29 copies) revealed a clear enrichment in C3H/HeJ relative to other ES cells, as shown in Figure 5A and Appendix A for ModEncode profiles, and two examples shown in Figure 5B. More specifically, 11 E14, 11 Bruce4 and 2 TT2 copies are associated with H3K4me3, while 11 common copies are in a permissive state exclusively in C3H/HeJ ES cells. The DNA hypomethylation of common IAP copies enriched for H3K4me3 only in C3H/HeJ ES cells is also specific to C3H/HeJ. Indeed, the full-length IAP copy (IAP #60), hypomethylated in C3H/HeJ ES cells, is also present in the 129 genome, but the LTR is highly methylated in J1 ES cells, along with flanking CpGs, as shown in Figure 5C. Another IAP solo LTR (IAP #44) is hypermethylated in TT2 ES cells while hypomethylated in C3H/HeJ ES cells, as shown in Figure 5C. Taken together, these observations indicate that a few C3H/HeJ IAPs are able to escape host silencing mechanisms.

### 3.6. Evidence for an Active “Master Copy” in the C3H Genome

The 7106 bp full-length H3K4me3-enriched IAP copy identified in the C3H/HeJ genome contains two nearly identical LTRs (LTR1) (one single mismatch), an intact primer binding site (PBS) and encodes functional IAPez retrotransposition machinery—namely full *gag, pro* and *pol* ORFs, as shown in Figure 6A and Appendix A. It is, therefore, likely to be a retrotranspositionally competent IAP or at least produce the retroviral proteins necessary for the retrotransposition of non-autonomous IAPs, such as 1Δ1. We searched for disruptions in known repressor sequences, such as the SHIN region (short heterochromatin inducing sequence), a 160 bp segment in the *gag* gene capable of triggering heterochromatin formation [49], although a direct role for this segment in suppressing IAP retrotransposition has not been tested. The full-length IAP copy contains a nearly intact SHIN region, despite being associated with H3K4me3, as shown in Figure 6B and Appendix A. 

No identical copies are found in the reference C57BL/6 genome. However, we were able to detect, by blast against the NCBI nucleotide database, the orthologous IAP copy in the 129/Sv genome, which was fortuitously sequenced as part of a X-inactivation study (GenBank: AJ421480.1) [43]. These orthologous IAP copies are extremely similar, as expected (99% identical), as shown in Appendix A, and there are no differences in key regulatory elements (5’LTR, PBS or SHIN region). When searching for similar IAP copies in NCBI databases, we detected two previously described IAP sequences derived from the mouse C3H/HeJ genome—*Atrn^mgL^* (GenBank: FJ854357.1) and Q14 (Genbank accession AB099818.1). The *Atrn^mgL^* mutant allele was caused by the insertion of a full-length IAP in an intron of the *Attractin* gene in C3H/HeJ [50]. The Q14 sequence, along with several 1Δ1 IAP copies, were isolated as novel somatic insertions in radiation-induced acute myeloid leukemia (AML) cell lines from C3H/He mice [51]. Our C3H full-length IAP copy is similar to both the *Atrn^mgL^* and Q14 elements (percentage identities of 98% and 93%, respectively), with most mismatches and indels observed within the LTRs. It is important to note that these IAP mutation sequences are different, with Q14 showing a 100 bp insertion in the 5’LTR, not present in the *Atrn^mgL^* nor the C3H/HeJ candidate master copy uncovered here. Therefore, in C3H/HeJ mice, more than one autonomous copy might be responsible for IAP retrotransposition. 

Interestingly, comparative analysis of the genomic region flanking this “master copy” in sequence assemblies of 16 available mouse strains [28] reveals it to be present in multiple strains in addition to C3H/HeJ, namely A/J, BALB/cJ, CBA/J, FVB/NJ, LP/J, 129S1 and NOD, although its full structure in these other strains is not known due to incomplete sequencing through the IAP. Notably, of the seven reported IAP germ line mutations that did not occur in C3H or in a C3H hybrid [8], one occurred in the old "Bussy stock" and, of the other six, five occurred in strains with the master copy or in a hybrid involving those strains—namely BALB, A-strain mice or CBA, as shown in Table 1 in [8]. Only one IAP mutation definitively occurred in a strain lacking this copy—namely DBA/2J [52,53]. Hence it is tempting to speculate that the presence of this “master” copy is important for germ line IAP retrotransposition, at least since the divergence of inbred strains. 

As discussed above, the 5’ LTR and internal region of this copy in 129-derived ES cells is heavily methylated, in contrast to its state in C3H/HeJ ES cells, as shown in Figure 5C. This observation suggests that, in the C3H background, this particular copy may be subject to less epigenetic repression. Interestingly, there is a decrease in methylation in the two CpGs surrounding the IAP insertion site of the full-length copy in C57BL/6 ES cells, suggesting that this insertion site might be propitious for permissive marks.

## 4. Conclusions

During mouse evolutionary history, the IAP family of retrotransposons has been very successful in colonizing the genome with thousands of copies. However, since the establishment of inbred strains, this colonization appears to have largely ceased, with C3H being the notable exception. Here we have shown that IAP insertions are accumulating more rapidly in C3H/HeJ compared to other sequenced strains, which is in accord with the fact that most documented germ line mutations due to new IAP insertions have occurred in C3H/HeJ or other C3H sub-strains [8]. Indeed, although our study used C3H/HeJ ES cells and genomic sequences, it is probable that other, less commonly used C3H sub-strains are also susceptible to IAP retrotransposition. We have also shown that a number of 1Δ1 IAP copies, specific to C3H/HeJ, have generated open chromatin in ES cells (unmethylated and marked with H3K4me3) and hence could be transcriptionally active. The much higher transcript levels of 1Δ1 elements in C3H/HeJ ES cells compared to other strains is likely due to the activity of these copies. Unfortunately, because of their near identity, we cannot determine which specific loci are actually transcriptionally active. Notably, despite the fact that hundreds of 1Δ1 sequences, as part of the reference C57BL/6 genome, are in Genbank, database searches using published sequences from mutation-causing 1Δ1 elements from C3H mice reveal that the only identical matches are sequences from 1Δ1 elements responsible for other mutation cases in C3H (unpublished observations). This suggests that many of these new insertions arise from just a few “transcriptionally hot” 1Δ1 loci in the C3H genome.

The reverse transcription of 1Δ1 mRNAs, no matter how abundant, and their subsequent reintegration into the genome requires a source of GAG and POL proteins from coding-competent elements. Here we have identified a single IAP copy with full open reading frames that is marked with H3K4me3 and harbors an unmethylated 5’ LTR in C3H/HeJ ES cells. While the epigenetic state of this copy in germ cells is unknown, this putative “master copy” could be the source of IAP proteins enabling the germ line retrotransposition of 1Δ1 elements. Notably, this copy is not unique to C3H/HeJ but, at least in 129/Ola ES cells (E14), is heavily methylated. While more work is necessary to investigate the epigenetic state of this copy in other strains, our data suggest that its activity in C3H/HeJ mice facilitates the high retrotransposition of IAP elements, specifically in this strain. It is important to stress that all the ES cells studied here were cultured on standard media rather than 2i media [23]. Therefore, the level of DNA methylation is likely higher than in inner cell mass cells [54,55], even though no reactivation of IAP copies has been observed in 2i conditions [55]. More important, H3K9me3 IAP deposition remains high in 2i-mouse ES cells [56]. Finally, we are likely missing other potentially permissive IAP copies due to the difficulty of mapping reads in repeated sequences. Although we took advantage of nearby sequences, some IAP copies inserted into other repeats were likely missed by our analysis.

In conclusion, it is probable that more than one factor contributes to high IAP “activity” in the C3H strain, as shown in Figure 6C. One factor could be a deficiency in the epigenetic repression of IAPs as a result of strain-specific differences in silencing pathways or proteins, such as those encoded by KRAB zinc finger genes [15,21,57], which are highly polymorphic between strains [28,58]. If such a deficiency exists, it must be subtle, or this trait would have been purged from the strain. However, even a subtle difference could, over time, allow for the accumulation of neutral IAP insertions in the C3H genome and an occasional mutation with phenotypic consequences. It appears that, at least some members of the 1Δ1 subfamily, in particular, escape transcriptional silencing in a C3H background, possibly because its partly deleted structure lacks sequences targeted by KRAB zinc finger proteins or other components of the silencing machinery [15,57]. A second factor could be the presence of the full-length coding-competent “master copy”, which is present in several strains but is likely subjected to less repression in C3H/HeJ. Future efforts to elucidate the molecular basis for the continued retrotransposition of IAP elements in this strain should shed light on our understanding of host defenses against such activity and of strategies by the IAP family to circumvent these defenses.

## Figures and Tables

**Figure 1 viruses-12-00783-f001:**
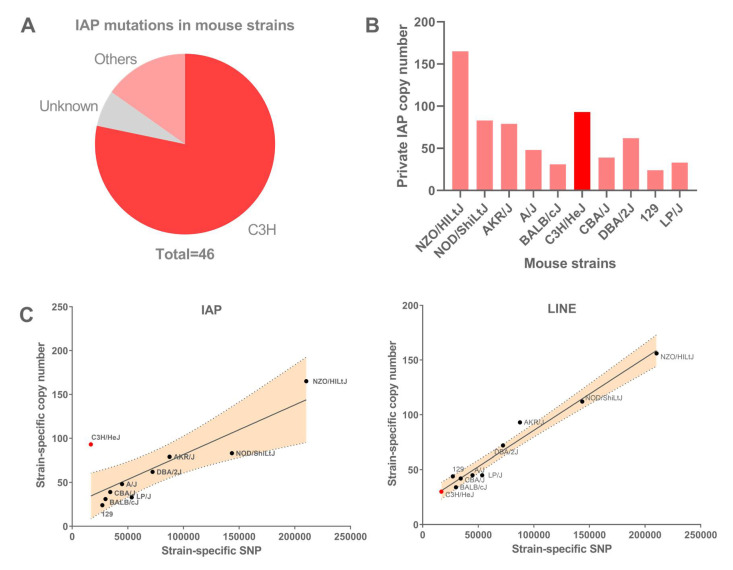
Accumulation of IAP elements in different mouse strains. (**A**) Distribution of IAP-induced mutations in mouse strains, adapted from [8], showing high occurrence in C3H- or C3H-derived strains. (**B**) Number of private IAP copies (i.e., present in one single strain), across the 12 studied strains from [26]. (**C**) Private IAP and LINE copy numbers, respectively, versus private SNPs, across mouse strains. C3H/HeJ accumulated significantly more IAP copies than the other mouse strains (linear regression with 95% confidence interval is depicted). Private copy numbers for 129/P2, 129/S1 and 129/S5 have been merged into a single 129 strain and the C3H/HeJ strain is depicted in red.

**Figure 2 viruses-12-00783-f002:**
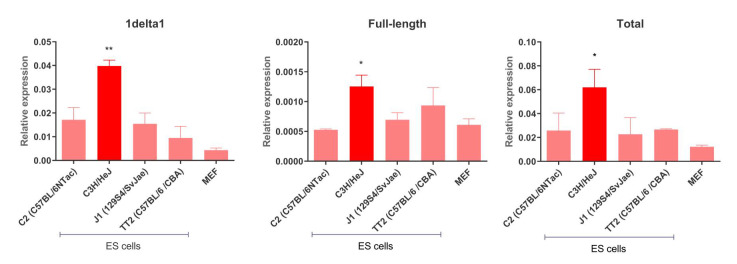
Steady-state levels of IAP transcripts in a panel of ES cells. Levels in MEFs are shown for comparison. Left panel shows 1Δ1 transcript levels, middle panel shows transcripts of full-length copies and right panel shows total IAP expression. Dark red bars highlight the C3H/HeJ IAP steady-state level. Anova with Dunnett’s pos-hoc test was performed and standard deviation is shown. * *p*-value < 0.05, ** *p*-value < 0.005.

**Figure 3 viruses-12-00783-f003:**
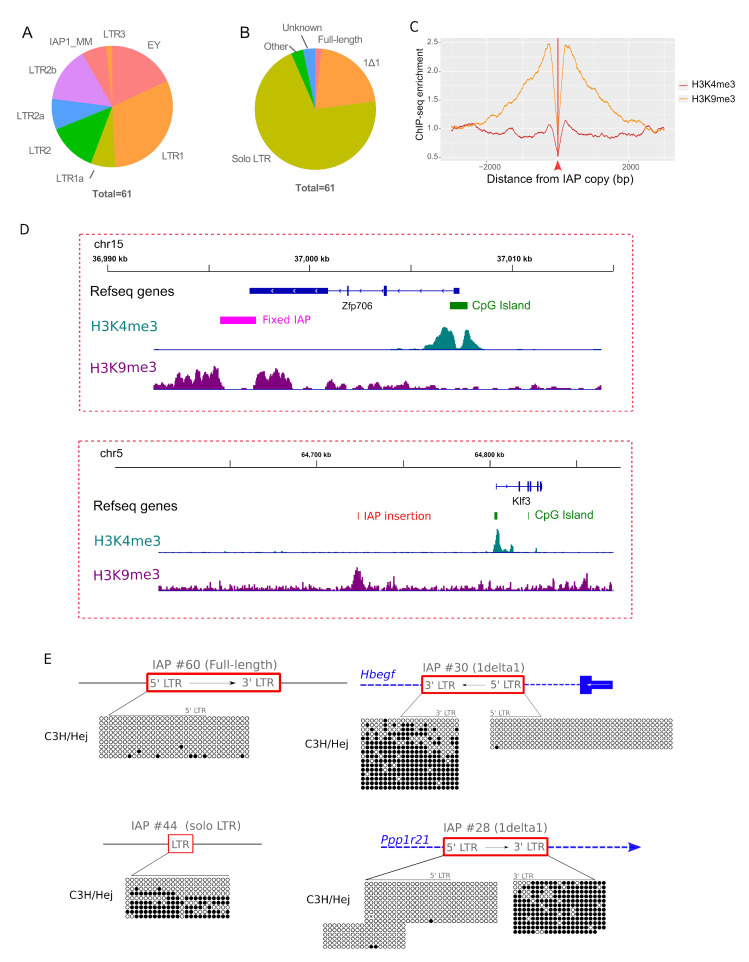
The C3H/HeJ ES cell genome has 61 IAP copies associated with H3K4me3. (**A**) Distribution of IAP subfamilies among the 61 H3K4me3-IAPs. Most IAP subfamilies are young (LTR1, lTR2 and EY). (**B**) Distribution of IAP types among the 61 H3K4me3-IAPs. (**C**) Average H3K4me3 and H3K9me3 in the flanking regions of all IAP insertions in the C3H/HeJ genome. The red arrow and line represents the IAP insertion site. (**D**) IGV genome browser view of IAP copies, either fixed (top panel) or C3H/HeJ specific (bottom panel), enriched in H3K9me3 in C3H/HeJ and lacking H3K4me3. For comparison, the closest H3K4me3-marked promoter is shown. (**E**) Bisulfite analysis of four H3K4me3-IAPs, including the full-length copy. Empty circles are unmethylated CpGs and full circles are methylated ones. An arrow depicts the IAP sense and a line above the CpG circles represents the CpG within the LTRs. In red, IAP insertions; in blue, genes; in black, intergenic regions.

**Figure 4 viruses-12-00783-f004:**
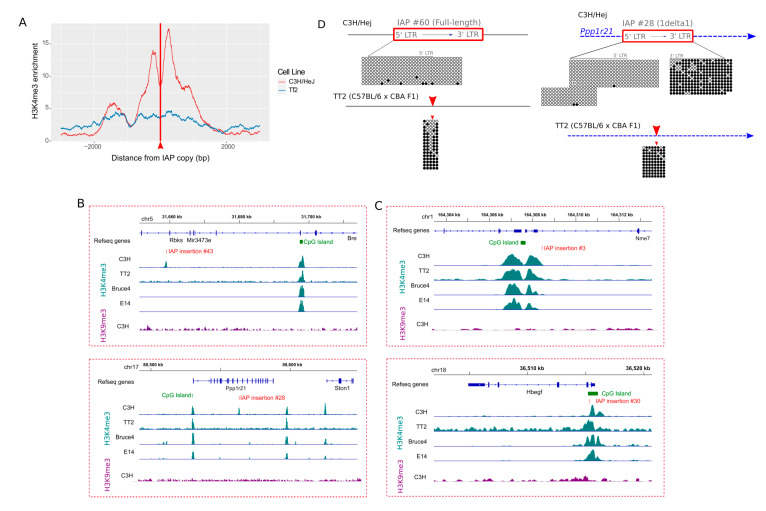
IAP copies are able to recruit H3K4me3 chromatin. (**A**) Average enrichment of H3K4me3 in TT2 (C57BL/6 × CBA) and C3H/HeJ ES cells in 27 regions flanking IAP copies present in C3H/HeJ and absent from C57BL/6 and CBA. The red arrow and line represents the IAP insertion site. (**B**,**C**) IGV genome browser view of IAP copies inserted in C3H/HeJ and absent from other mouse strains. IAP copies #43 and #28 are able to recruit H3K4me3 (**B**), while the empty sites in TT2, Bruce4 and E14 show no H3K4me3 enrichment. For comparison, the closest H3K4me3-marked promoter is shown. IAP copies #3 and #30, present only in C3H/HeJ, are inserted within H3K4me3-marked regions (**C**), as seen in the empty sites of TT2, Bruce4 and E14. (**D**) Bisulfite analysis of two empty sites in TT2 cells corresponding in C3H/HeJ to the full-length insertion (IAP copy #60) and a 1Δ1 insertion in the Ppp1r21 gene (IAP copy #28). The methylation profile in C3H/HeJ cells, presented in Figure 3E, is shown here for comparison. Empty circles are unmethylated CpGs, full circles are methylated ones. The arrow inside the IAP insertion box depicts the IAP direction and the line above the CpG circles represents the CpGs within the LTRs. Red boxes are IAP insertions; in blue, genes; in black, intergenic regions. The red arrow head represents the C3H/HeJ IAP insertion site in the empty C57BL/6 genome.

**Figure 5 viruses-12-00783-f005:**
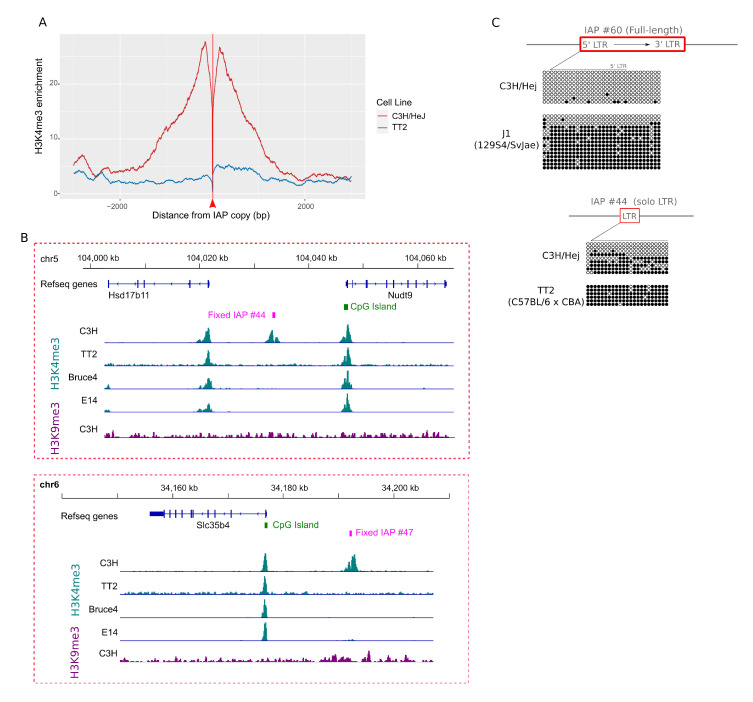
(**A**) Average profile of H3K4me3 at the flanking sites of 29 IAP copies common to C3H/HeJ, 129 and C57BL/6 in C3H/HeJ and TT2 ES cells. The red arrow and line represents IAP insertion site. (**B**) IGV genome browser view of two common copies (IAP #44 and # 47) showing H3K4me3 only in the C3H/HeJ strain. One can observe the enrichment of H3K4me3 at a nearby CpG island promoter in all four ES cells. (**C**) Bisulfite analysis of two H3K4me3-IAPs in J1 (129S4/SvJae) and TT2 (C57BL/6 × CBA F1) ES cells, including the full-length copy (IAP #60) and a solo LTR (IAP #44). The methylation profile in C3H/HeJ cells, presented in Figure 3E, is shown again for comparison. Empty circles are unmethylated CpGs, full circles are methylated ones. The arrow depicts the IAP direction and the line above the CpG circles represents the CpG within the LTRs. In red, IAP insertions; in black, intergenic regions. LTR: long terminal repeat.

**Figure 6 viruses-12-00783-f006:**
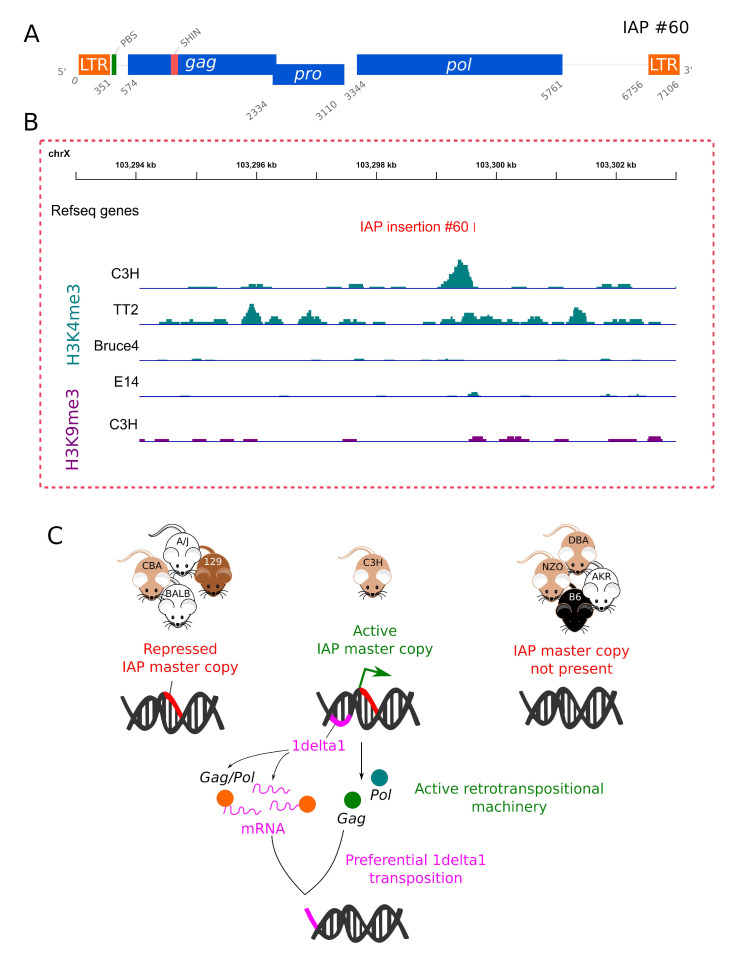
The IAP master copy. (**A**) Diagram of the potential IAP master copy (IAP #60) with intact open reading frames coding gag, pro and pol, an intact PBS region (green) and the presence of the repressive SHIN region (pink). (**B**) IGV genome browser view of the master copy, which is present in the C3H/HeJ genome along with the 129 genome (E14 ES cells), but is absent from the C57BL/6 genome (Bruce4 and TT2 ES cells). H3K4me3 enrichment is only observed in the C3H/HeJ cells. The position of the insertional polymorphic master copy is depicted in red. (**C**) Model of IAP transposition in C3H mice. The C3H master copy is present in several mouse strains, but its permissive state may be specific to C3H mice. The synthesis of functional retrotransposition machinery is, therefore, possible in C3H, and allows 1Δ1 transposition in trans. Colored circles represent proteins produced by full-length IAP (GAG and POL) and a GAG-POL fusion protein produced by 1Δ1 IAP copies, which facilitates their retrotransposition.

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
