# Peer review of "Inter-Strain Epigenomic Profiling Reveals a Candidate IAP Master Copy in C3H Mice"

_viruses, 2020, doi:10.3390/v12070783_

Round 1

Reviewer 1 Report

IAP elements are a frequent class of endogenous retroviruses in the mouse genome. In most cell lines, expression of these elements is restricted by a series of host factors that establish a transcriptionally repressive chromatin state. Here, the authors characterize a mouse substrain with unusually high IAP activity in ES cells. They identify a number of presumably active IAP insertions, which are only partially shared with other mouse strains. Furthermore they discover a potentially active full length IAP insertion in C3H mice, which could serve as source for IAP protein production, needed for retrotransposition of mutated IAP copies.

Although, direct evidence is missing for the role of the "master copy" to drive IAP transposition, the manuscript is still interesting for the aspects of differential transcriptional regulation of IAP copies between mouse strains. For example, it seems that in C3H mice, IAP copies with conserved silencing recruitment sequences could be transcriptionally active.

The following points need to be revised in order to increase readability of the mansucript.

1. Cumulative coverage plots of IAP insertions (Fig 1C, ...) are probably aligned to the insertion site and should lack the signal in the insertion; so these plots should have no signal in the center. All these plots need to be corrected.

2. A consistent nomenclature of IAP insertions should be used in text and figures. It is very hard to follow in text vs figures, which specific IAP insertion is meant and whether this insertion is specific to C3H or present also in other strains. As mainly insertions from Table S2 are discussed, it would be good to add an identifier to this table and use this identifyer in both text and figures.

3. The definition of potentially active IAPs is based on H3K4me3 spreading from the presumably active LTR. It is not fully clear to me if this selection includes insertions which contain H3K4me3 from a nearby promoter, as e.g. in Fig. 4C. Since H3K4me3 does not seem to originate from the LTR but rather declines toward the LTR one would rather classify these elements as inactive.

4. For active IAP elements it would be good to include H3K9me3 in the analysis and the screenshots to know if they exclude each other on active vs. inactive copies.

5. All bisulfite data need to be better annotated to increase readability (specify IAP insertion, and mouse strain in the figure); comparisons between mouse strains are mentioned in the text (e.g. lines 319-321) but not shown in the figure.

6. It is interesting that the "master copy" is active in C3H, but inactive in other strains, despite having conserved 5'UTR and SHIN regions. This statement would profit from showing alignments to document the statement. Finally, it would be very interesting to clarify whether C3H cells are capable of silencing reporters that can easiliy be inactivated in ES cells from other strains.

Author Response

We thank the reviewer for the comments and have addressed them as follows:

  1. We believe the reviewer is referring to figure 3C and subsequent figures and are sorry this wasn’t clear. The coverage plots were made using deeptools, plotprofile function. The coordinates used for the plots were indeed the insertion sites of IAPs (either 1 bp for an insertion site absent from the reference B6 genome, or the full insertion if the copy was present in B6). All insertions (1bp or larger) were scaled to 1bp, and this is indeed represented by the enrichment at the middle of the plots (0). Therefore, the plots are showing the flanking regions, but the 0 shows the enrichment in the IAP (if inserted in B6) or in the 1bp insertion site. We have added an arrow and a line at the insertion site, to explicitly show the IAP insertion site and remove the ChIP-seq enrichment (see revised Figures 3C, 4A, 5A and S4, S5 and S6).

2. This is a good point and we thank the reviewer for the suggestion. We added an IAP identifier in Table S2 (first column) and have explained this in section 3.3. We have re-annotated all figures and revised the text to clarify IAP copy discussion using the identifier numbers.

3. Indeed, the “active IAP copies” do include a few insertions that contain H3K4me3 from a nearby promoter (see IAP #3, IAP #30 for instance, and figure 4C) and we discuss this in the text. We have previously shown that IAP copies close to H3K4me3 gene TSSs can also be marked by H3K4me3 and present an active promoter (see Rebollo, Miceli-Royer et al. Genome Biology 2012). Therefore, even if an IAP is inserted into an H3K4me3 domain due to an active gene TSS, it can activate its own promoter.

4. As suggested by the reviewer, we added the C3H/HeJ H3K9me3 track to all genome browser views (Figures 4B and C, 5B and 6B). We also added two new genome browser screenshots to Figure 3D to show H3K9me3 enrichment in a fixed IAP and a C3H-only IAP copy that lack H3K4me3 (so they are not in Table S2). Finally, we included an average plot of H3K9me3 and H3K4me3 enrichment in all copies of Table S2, i.e., all copies that have H3K4me3 in C3H/HeJ (new Figure S4). (The original Figures S4 and S5 have been renamed S5 and S6).

5. We agree with the reviewer that the bisulfite data presentation was confusing and have annotated the copies in all bisulfite figures with strain information and IAP identifier numbers. As well, we explicitly added the bisulfite comparisons in the figures.

6. We have annotated the IAP #60 (full length master copy) in the Supplementary data to include all the significant regions (LTR, ORFs, PBS, SHIN), and have added an alignment of the master copy in C3H with the orthologous copy in the 129 genome in Supplementary data 1. 

Indeed, it would be very interesting to compare reporter silencing between C3H ES Cells and other strains, but unfortunately this goes beyond the scope of the manuscript and we hope other researchers will be able to address this question.

Reviewer 2 Report

This manuscript reports the identification of a subset of IAP elements that are transcriptionally active and also present in one strain of the mouse, C3H/HeJ, and also a potential master copy that is responsible for the amplification of this C3H-specific IAP elements.  The authors employed a series of comparative genomic and bioinformatic approaches to identify this particular subset of IAP elements.  They further revealed that a large portion of IAP insertions reported recently are derived from this subfamily.  The authors also analyzed the transcription activity and epigenetic profiles of this subfamily to further confirm the retrotransposition activity of this subfamily in the C3H/HeJ stain.  Finally, the authors have identified one full-length IAP from X chromosome and provided several lines of evidence suggesting that this IAP may be a master for the recent amplification of this IAP subfamily. 

The main finding of the current manuscript is the identification of a potential master IAP element that may have been responsible for the retrotransposition of IAP elements in one strain of the mouse.  This is particularly significant, since the current technology involving molecular and mouse genetics may provide potentially a large amount of previously unknown aspect of retrotransposition biology through dissecting this particular IAP.

Nevertheless, there are a couple of minor criticisms for this manuscript.

First, although hinted indirectly at several spots in the introduction and results sections, the authors need to emphasize explicitly the fact that the amplification of this IAP subfamily has occurred in very recent times, most likely after the establishment of the majority of individual mouse strains.  This clarification should strengthen the authors’ claim, given the fact that the genomes of mouse strains are overall mosaics due to the inter-breeding and subsequent mixing of individual strains at the early stage of mouse genetics.

Second, the term full-length in Fig. 1 is not well understood in relation to 1D1, because it appears so sudden without any explanation.  So, the authors need to explain what is full-length versus 1D1 IAP.  This is also the case for the small domain SHIN in Fig. 6.  The authors need to explain the significance of this domain in terms of the retrotransposition activity of IAP for the general audience.

Author Response

We thank the reviewer for the comments and have addressed them as follows.

Response to first comment:

Actually, it is clear that IAPs, including the 1delta1 subfamily, amplified prior to the divergence of inbred strains since most copies are indeed shared between more than one strain. This is shown in figure 1C. For example, even in the most divergent strain (NZO), only ~160 IAPs (out of several thousand) are specific to that strain. Our point in the manuscript is that, since the strains were established, the only significant further accumulation of IAPs has occurred in C3H. We had pointed this out in the first two sentences of “concluding remarks” but have added this additional sentence of clarification at the end of section 3.1: "However, it should be noted that the vast majority of the thousands of IAP copies in the genomes of inbred mice inserted prior to separation of the strains". 

Response to second comment:

We believe the reviewer is referring to figure 2, rather than figure 1. We added a sentence clarifying what is meant by “full length” in section 3.2 (Namely an intact gag and pol gene). We also added the full name of the SHIN region in section 3.6 as well as a phrase indicating that a direct role for this region in suppressing retrotransposition has not been tested.

Round 2

Reviewer 1 Report

The authors have satisfactorily addressed my concerns.